# Direct identification of reaction sites on ferrihydrite

Jean-François Boily [1✉] & Xiaowei Song [2]

Hydroxyl groups are the cornerstone species driving catalytic reactions on mineral nanoparticles of Earth's crust, water, and atmosphere. Here we directly identify populations of these groups on ferrihydrite, a key yet misunderstood iron oxyhydroxide nanomineral in natural sciences. This is achieved by resolving an enigmatic set of vibrational spectroscopic signatures of reactive hydroxo groups and chemisorbed water molecules embedded in specific chemical environments. We assist these findings by exploring a vast array of configurations of computer-generated nanoparticles. We find that these groups are mainly disposed along rows at edges of sheets of iron octahedra. Molecular dynamics of nanoparticles as large as 10 nm show that the most reactive surface hydroxo groups are predominantly free, yet are hydrogen bond acceptors in an intricate network formed with less reactive groups. The resolved vibrational spectroscopic signatures open new possibilities for tracking catalytic reactions on ferrihydrite, directly from the unique viewpoint of its reactive hydroxyl groups.

[1] Department of Chemistry, Umeå University, SE-901 87 Umeå, Sweden. [2] R&D, IKEA of Sweden AB, SE-343 34 Älmhult, Sweden.
✉email: jean-francois.boily@umu.se

Ferrihydrite (Fh) is a puzzling nanosized mineral with a 50-year-old history of debates over its structure, composition and formation mechanisms[1–20]. Fh nanoparticles display large concentrations of reactive hydroxyl (OH) sites responsible for scavenging and transforming contaminants[21,22] and nutrients[23,24], in natural and industrial waters, and in terrestrial environments. Interactions with these OH groups can even alter the conversion pathways of Fh to other iron oxides[25–30], and permanently occlude these compounds in the crystalline structure of these minerals. Still, our lack of knowledge on the composition and structure of Fh nanoparticle surfaces continues to challenge understanding reaction mechanisms that Fh hosts in nature and technology[31–34].

Understanding the coordination environments and spatial dispositions of reactive OH groups is the cornerstone for predicting mineral reactivity. Resolving this enigma for ferrihydrite is even more important since it can guide similar efforts for poorly defined materials currently facing similar challenges. Binding sites on (oxy)(hydr)oxide minerals consist of oxo groups bound to one ($-O$), two ($\mu-O$), or three ($\mu_3-O$) underlying iron atoms (Fig. 1). We can explore these populations at crystalline material surfaces by inspecting the underlying core structure and the main crystal habits of mineral nanoparticles. Applying this approach to Fh is however problematic given its more variable composition, degrees of crystallinity and defects, and unresolved crystal habits, all of which is intrinsically tied to the conditions (e.g. oxidation/hydrolysis rates, pH and temperature) in which nanoparticles formed. Despite these uncertainties — and opposing views (Drits[7,8,10,13,17], Michel[6,9,11,12,18] or hybrid[14] models) over Fh structure — Hiemstra[35–38] explored an approach for rethinking the surface chemistry of Fh by identifying OH populations on crystallographic faces of idealised nanoparticles. The model

envisions crystalline Fh nanoparticles composed of (1) a defect-free core consisting of the low OH-bearing structure of Michel et al[6]. ($Fe_5O_8H$), and (2) crystallographically-oriented surfaces of greater $OH/H_2O$ loadings ($Fe_5O_8H + n\ H_2O$) but depleted in Fe2 octahedral and Fe3 tetrahedral sites (Fig. 1). Such nanoparticles expose high densities of surface $-OH$ groups, and can reproduce the experimentally lower mass densities that cannot be predicted using the core crystallographic structure alone. Still, considering that real Fh nanoparticles have variable degrees of crystallinity, and that they are not terminated by idealised crystallographic faces, direct experimental identification of surface OH groups is strongly needed to advance our understanding of Fh surface chemistry.

This study addresses this need by directly identifying OH groups on synthetic Fh nanoparticles (Supplementary Fig. 2). Inspired by the vibrational spectroscopy work of the catalytic alumina literature[39,40] and by earlier efforts on iron oxides[41–46], our approach relies on the detection of an elusive set of O–H stretching bands that are phenomenally sensitive to the coordination environments of surface OH groups[43,44]. Although Russell[42] did publish over 40 years ago the first evidence for Fh surface OH groups, and these findings were later substantiated by Hausner et al.[47], these sites have never been clearly identified. However, we have in the recent years[45,46] established new measures to identify OH groups on crystalline iron (oxyhydr)oxides (Supplementary Fig. 1) and, in this study, we implement these new capabilities to directly identify OH groups on Fh. We support our findings with an exhaustive theoretical analysis of OH populations on Fh nanoparticles of spheroidal morphology that could be more representative of real materials. This becomes important as spherical cuts of crystallographic structures are likely to favour O(H) populations of lower coordination with underlying Fe sites than on flat crystalline surfaces, which have so far been considered[35–38]. We also accomplished the field's first set of molecular dynamics simulations of single Fh nanoparticles with realistic protonation states. These simulations provide novel insight into the structure and populations of hydrogen bonds developed in the $OH/H_2O$-rich surfaces of Fh. Our work now provides direct evidence for the dominance of reactive $-OH$ groups embedded within a hydrogen bonding network built from donating $\mu-OH$ groups. As spectral signatures of these groups are highly sensitive to changes in coordination environments, our work provides a path for exploring Fh-driven catalytic reactions under a new light, namely from the unique viewpoint of OH groups.

## Results and discussion

**Direct detection of OH populations**. We resolved OH populations of synthetic Fh nanoparticles as Fe-bound surface OH and $H_2O$ groups (3620–3690 $cm^{-1}$; bending region in Supplementary Fig. 3) locked in very specific chemical environments, and as core OH groups (3395 $cm^{-1}$) involved in a broader network of hydrogen bonds (Fig. 2a, b). X-ray photoelectron spectroscopy of strongly dehydrated Fh nanoparticles in ultra-high vacuum (Fig. 2c) revealed nearly equivalent concentrations of OH and O groups, and an overall composition of $FeO_{0.71}(OH)_{0.76} \cdot 0.05\ H_2O$ ($Fe_5O_{7.6}H_{4.3}$; Fig. S2, Table S1). We will later show that this composition is consistent with the concept of a hydroxyl-rich surface region covering a hydroxyl-poor Fh core[35–37].

We identified surface OH groups by comparing their spectral response with our previous assignments for crystalline FeOOH surfaces (Fig. 2d, Supplementary Fig. 1). This comparison shows that Fh generates flagship bands of poorly hydrogen bonded $-OH$ groups (3667 $cm^{-1}$), chemisorbed water molecules ($-OH_2$; ~3690 $cm^{-1}$) and doubly-coordinated hydroxyl groups ($\mu-OH$;

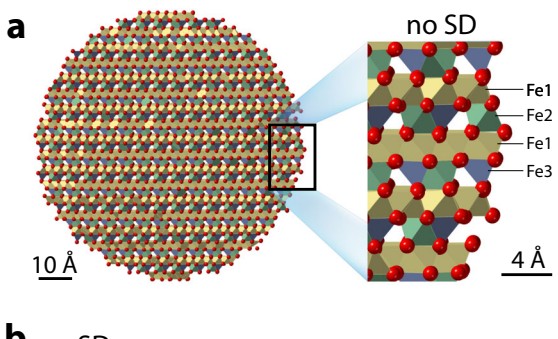

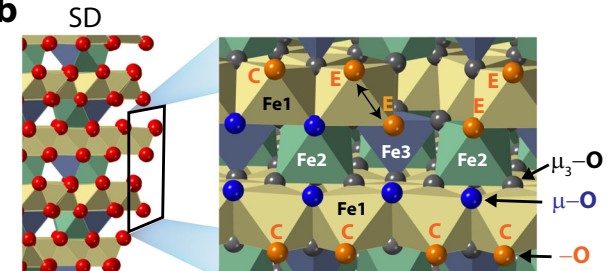

**Fig. 1 Schematic representation of oxo groups at Fh surfaces. a** Spherical Fh nanoparticle (8 nm diameter) cut from the structure of Michel et al.[6] with a starting composition of $Fe_5O_8H$ ($FeO_{1.4}(OH)_{0.2}$). A close-up of the selected area shows layers of octahedral (Fe1, Fe2) and tetrahedral (Fe3) iron atoms without (no SD) surface depletion of Fe2 and Fe3 sites. **b** Same close-up of selected area in Fig. 1a but with (SD) surface depletion of Fe2 and Fe3 sites. The face view of the selected area shows positions of Fe and O sites, including singly coordinated corner ('C') and edge ('E') oxo $-O$, and doubly-coordinated $\mu-O$. Triply-coordinated $\mu_3-O$ groups chiefly belong to the core.

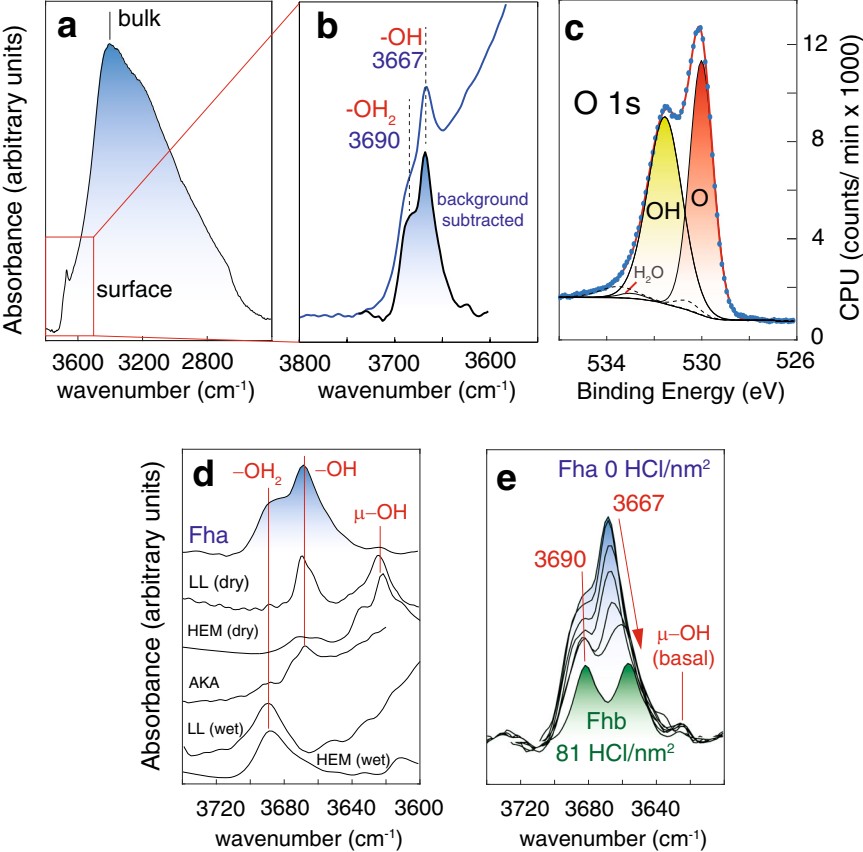

**Fig. 2 Spectral evidence for OH groups at Fh surfaces.** Surface OH group populations resolved by vibration (**a**, **b**, **d**, **e**) and X-ray photoelectron (**c**) spectroscopy. **a** Vibrational spectra of dry Fh under $N_2(g)$ revealed core OH and surface $-OH$, $\mu-OH$ and $-OH_2$. **b** Surface OH groups are on the high energy portion of the O–H stretching region (red square of **a**), and resolved further following removal of background contributions from core OH groups. **c** X-ray photoelectron spectroscopy of the O 1s region revealing a surface composition of $FeO_{0.71}(OH)_{0.76} \cdot 0.05 H_2O$ for Fh under ultra-high vacuum. Low intensity dashed lines are O species of strongly resilient trace carbonate and carboxyl contaminants despite the efforts made in isolating this material from the atmosphere (Table S1). **d** Collections of narrow surface OH bands reflect co-existing OH groups of contrasting coordination environments. Surface OH bands of Fh are highly comparable to key bands of akaganéite (AKA; β-FeOOH), lepidocrocite (LL; γ-FeOOH) and haematite (HEM; α-Fe$_2$O$_3$) under dry and wet conditions (~4 H$_2$O monolayers)[45,46,48]. See Supplementary Fig. 2 for more information. **e** Surface OH groups respond to proton loadings, here showing the preferential consumption of $-OH$ groups (3667 cm$^{-1}$).

3620 cm$^{-1}$). Triply-coordinated $\mu_3-OH$ sites were not detected in the expected spectral range where they occur in other crystalline iron (oxyhydr)oxide[43,45,48] minerals (cf. Fig. 2a in the 3491–3578 cm$^{-1}$ region, and Supplementary Fig. 1 for reference spectra). While strong overlap with core OH bands or hydrogen bonding may have hindered spectral resolution, another is possibility, also raised by Hiemstra[35–37], is that $\mu_3-OH$ sites are of lower density on Fh surfaces. We will return to this in our theoretical analyses of OH populations.

Our previous work showed that $-OH$ groups responsible for the 3667 cm$^{-1}$ band are disposed along rows on dominant faces of FeOOH minerals (Supplementary Fig. 1)[43,48]. Such rows are also exposed at the edges of sheets of Fe1 octahedra in our idealised representation of Fh (Fig. 1), and co-exist with chemisorbed water ($-OH_2$) or bare Fe Lewis acid sites in acid-neutral and charge-neutral surfaces. We note that rows with greater populations of inter-site hydrogen bonds, such as in the mineral goethite (α-FeOOH)[43], generate a band downshifted by 6 cm$^{-1}$, and are not present in Fh. This difference is highly significant as these bands are phenomenally responsive to the slightest changes in hydrogen bond strength, and to the tune of ~150 cm$^{-1}$ per pm change in O–H bond length[44]. This 3667 cm$^{-1}$ band is therefore key evidence for a dominance of

Fh $-OH$ groups of relatively lower hydrogen bonded environments than on goethite.

To illustrate the sensitivity of $-OH$ groups to their bonding environments, we altered their speciation by exposing Fh to HCl (Fig. 2e). The protonation reactions that followed triggered the same spectroscopic response that we have resolved in our reference crystalline FeOOH minerals[43,48], and can be appreciated by the loss of our 3667 cm$^{-1}$ band and its shift to lower frequencies. These changes signal that new hydrogen bonds were formed between unreacted $-OH$ groups and $-OH_2$ sites along the rows we have previously mentioned. It is these chemisorbed water molecules that contribute to our second flagship band at ~3690 cm$^{-1}$. The high O–H stretching frequency of these water molecules is from an entirely isolated O–H bond from the hydrogen bond network of the Fh surface, and we note that we have previously detected such species on iron (oxyhydr)oxide[43,45,48] nanominerals. In this previous work, we showed that while a fraction is always resilient to outgassing under dry gases, water can be removed by prolonged exposure to ultra high vacuum or heat. We will come back to this point when looking at the thermal stability of Fh.

The band for $\mu-OH$ was only of small intensity, yet its appearance is of high value for understanding the surface

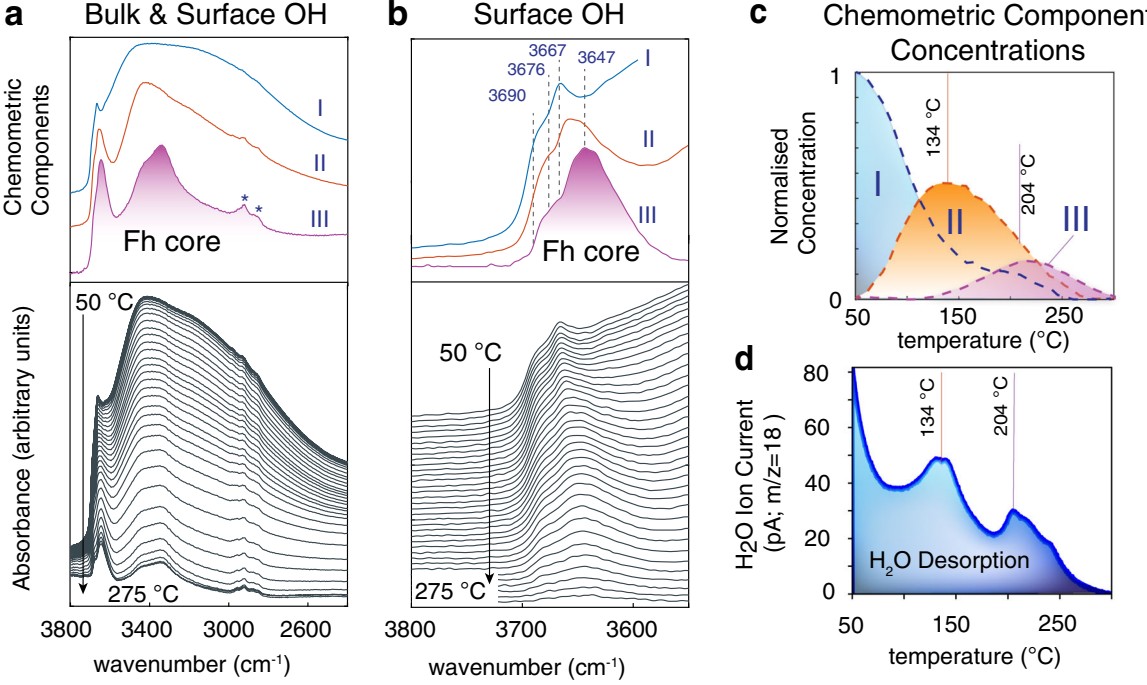

**Fig. 3 Temperature-resolved populations of surface and bulk OH groups in Fh temperature-programmed desorption of Fh at a rate of 10 °C/min at ~0.3 Pa. Similar results were obtained in N₂(g).** Broad (**a**) and narrow (**b**) O–H stretching region respectively showing core and surface dehydroxylation. A chemometric analysis of the data extracted spectral components I, II and III. The asterisk in **a** is from adventitious organic contamination. **c** Concentration profiles of chemometric components. **d** Quadrupole mass spectrometric detection of evolved water ($m/z = 18$, ion current measured in pA). Water release profiles align with the dehydration events of component II peaking at 134 °C and of component III at 204 °C. The core is nearly completely dehydroxylated at 375 °C (Supplementary Fig. 4).

chemistry of Fh, as it is a well-known signal for groups on the basal faces of iron (oxyhydr)oxides (Supplementary Fig. 1)[43,45]. This suggests that, although generally of spheroidal morphology, Fh nanoparticles could have developed a minor basal face exposing neutrally charged and proton inactive μ−OH groups[43,45]. Our previous work on this face shows that these groups are embedded in a highly regular hydrogen bonding environment responsible for the narrow 3620 cm⁻¹ band. The other μ−OH groups of the dominant spheroidal surface are, in contrast, not discernable as they are involved in a broad network of hydrogen bonds. We have arrived to the same conclusion through our work on crystalline iron (oxyhydr)oxides[43,45], and will support this further in molecular simulations in the last section of this study.

We can validate these findings further by inspecting the thermal response of these spectral signatures at temperatures well below the conversion temperature (e.g. 380 °C as in Xu et al.[12]) of Fh to haematite (Figs. 3 and S3)[2,12]. Exposing Fh nanoparticles to a heating gradient in vacuo progressively removed all water, as well as surface and core OH groups (Fig. 3a, b). This loss correlates first with the continual evacuation of water below ~100 °C leading to dehydration events at ~134 °C and at ~204 °C (Fig. 3d). These two events removed the great majority bound water (see Supplementary Fig. 3 for the bending mode of water), chemisorbed water (3690 cm⁻¹) and −OH groups (3667 cm⁻¹). These events produced a high temperature Fh core exposing the most heat-resistant core OH and surface μ-OH groups. Using a chemometric analysis[49] we show that the concentration profile (components III of Fig. 3c) of the Fh core directly aligns with the dehydration events at ~134 °C and ~204 °C, both marking transition temperatures between dominant Fh hydration states. The spectral signature of this core is also dominated by surface μ−OH (3643 cm⁻¹) and core OH (3395 cm⁻¹) groups.

Fh nanoparticles thus chiefly exposed isolated surface −OH and hydrogen-bonded μ−OH groups of comparable bond strength in our reference crystalline iron (oxyhydr)oxide minerals (Supplementary Fig. 1)[43,48]. The μ−OH groups become the sole surface species at high temperatures after removal of chemisorbed water and −OH sites. The spectra offer, however, no clear evidence for the presence of μ₃−OH groups at the surface. Finally, we note that no evidence could be found for the exposure of OH groups bound to a Fe3 (tetrahedral Fe) site. Drawing from the catalytic alumina literature[39,40], where OH groups bound to tetrahedrally-coordinated $Al^{3+}$ are well documented, we would have expected a band overlapping with that of the chemisorbed water. However, as this band shifted congruently with gradients in proton loadings (Fig. 2e) and temperature (Fig. 3b), we have no evidence for such a site on Fh surfaces.

By resolving the complex vibrational spectroscopic signals of Fh, we here offer new possibilities for tracking the catalytic reactivity of Fh from the unique viewpoint of its OH groups. We can, of instance, begin resolving the binding of critically important gases (e.g. $CO_2$, $CH_4$, $NO_x$, $SO_x$) on Fh. As these spectroscopic signals also persist under atmospheric humidity[47,50], we can explore interfacial reactions and phase transformation in nanometrically thick water films formed at Fh surfaces. In the following sections, we bridge these findings with new theoretical analyses and molecular simulations underscoring the importance of −OH and μ−OH groups on the surface chemistry of Fh nanoparticles.

**Populations and spatial dispositions.** We support our experimental findings by identifying OH groups on simulated Fh nanoparticles of spheroidal morphology. Spheroidal nanoparticles, with no or little defined crystallographic faces, are not only more representative of real Fh but are also expected to

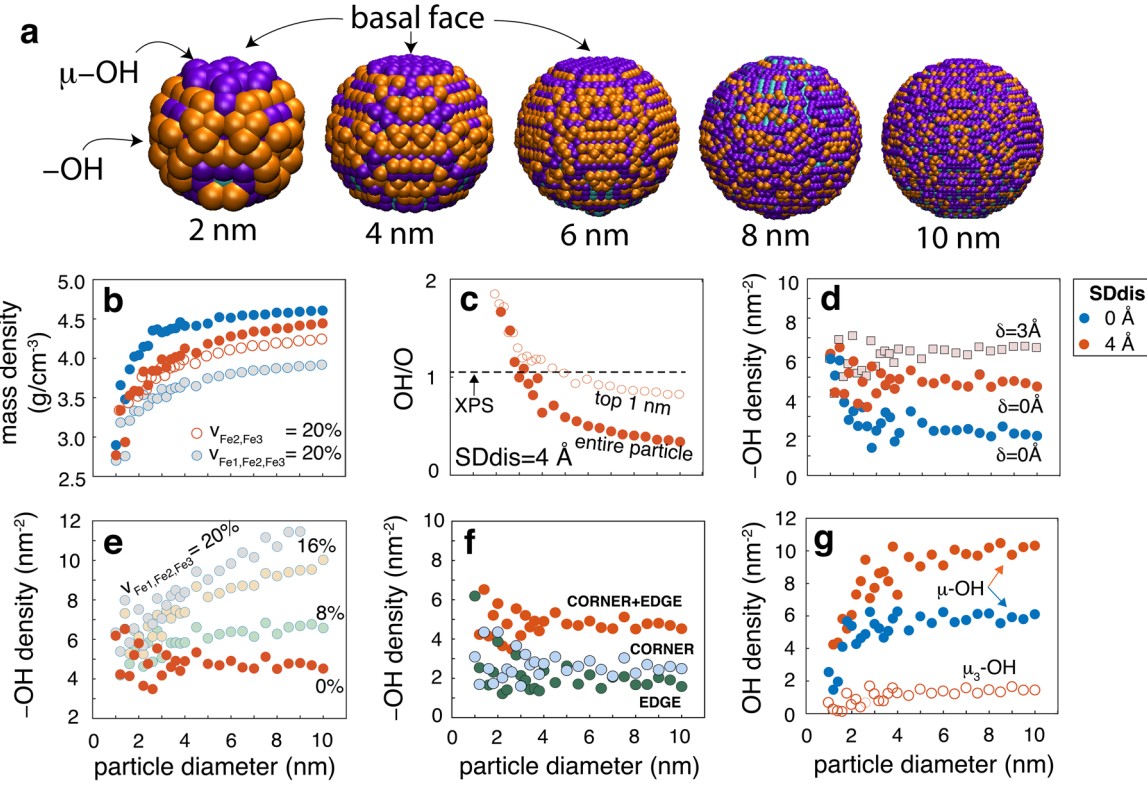

**Fig. 4 OH populations and mass density in simulated Fh nanoparticles. a** Simulated Fh nanoparticles at selected diameters, and their terminating hydroxo groups (orange = −OH; purple = μ−OH; turquoise = μ₃−OH). The μ−OH groups of the basal faces are more clearly expressed in particles up to 6 nm. **b** Mass densities at given SDdis values and Fe vacancies. **c** OH/O ratios at SDdis=4 Å for entire and top 1 nm of the particles. The dashed horizontal line marks the OH/O ratio of ~1.1 obtained by X-ray photoelectron spectroscopy (XPS). **d-g** Size dependence of OH density for **d** −OH groups affected by surface depletion depth (SDdis) and atomic displacement of δ=3 Å. **e** −OH groups affected by Fe vacancies at SDdis=4 Å. **f** Breakdown of corner and edge −OH groups (Fig. 1) at SDdis=4 Å (*n.b.* two "edge −OH groups" form one edge bidentate site). **g** μ−OH and μ₃−O densities at given values of SDdis. See Supplementary Figs. 5–7 for supporting results.

expose greater proportions oxygens of lower Fe coordination number than on crystallographically flat surfaces. To explore this idea, nanoparticles were generated from a spherical cut of the crystalline structure of Michel et al.[6], and were modified for various surface depletion depths of Fe2 and Fe3 sites (0–6 Å), atomic displacements (0–4 Å) and Fe site vacancies ($v_{Fe}$). All broken Fe–O bonds were healed and protonation levels were set to −OH and μ-OH, while those of μ₃−O(H) sites were set to levels predicted from the core structure. Finally, all nanoparticles were made charge-neutral by protonation of a required number of randomly selected −OH sites to −OH₂. Inspection of 280 computer-generated nanoparticles reveals a dominance of −OH/−OH₂ and μ−OH groups but only low densities of μ₃−O(H) sites at Fh surfaces (Fig. 4; Supplementary Figs. 5–7 for full set of results). These low μ₃−O(H) densities align with our experimental results, and again suggest that the spherical Fh surface should be dominated by −OH/−OH₂ and μ−OH groups.

The more relevant nanoparticles to be considered in our analysis are those whose mass densities (Fig. 4b; corresponding molar mass in Supplementary Fig. 6) are closer to experimental values (e.g. ~3.4 g/cm⁻³ for ~2–3 nm particles[11]). In accordance with the work of Hiemstra[35–37] on crystalline Fh nanoparticles, we find that depleting surface Fe2 and Fe3 sites is a highly effective means for reproducing experimental mass densities. A depletion depth of no more than 4 Å from the average positions of the top O atoms is needed to obtain such values. We can also justify introducing up to ~20% additional vacancies Fe2 and Fe3 vacancies ($v_{Fe2,Fe3}$) throughout the Fh core (Figs. 4b and S7). We can, however, introduce only small levels of vacancies at Fe1 sites,

given X-ray total scattering work[51] pointing to nearly fully occupied octahedral sheets.

Another experimental constraint to consider prior discussing OH populations, is the OH/O ratio of ~1.1 obtained by X-ray photoelectron spectroscopy of our strongly dehydrated Fh nanoparticles in ultra-high vacuum. The OH/O ratios retrieved from our theoretical considerations (Fig. 4c) were strongly size-dependent as they reflect contrasting proportions of the OH-rich surface and OH-poor core. In this context, only the ratio at ~3 nm aligns with the ~1.1 ratio (Fig. 2c). We can also obtain this ratio for larger particles sizes by consideration of the uppermost region of the particles and by contributions of non-stoichiometric core OH groups generated by vacancies.

Our search within particles with a range of reasonable mass densities shows that surface depletion of Fe2 and Fe3 sites is the dominant factor altering −OH densities on Fh (Fig. 4d). We find that a surface depletion down to 4 Å from the surface more than doubled −OH sites densities, yet depletions at greater depths had no additional impact. Our simulations also show that atomic displacements, although important in the elucidation of core structure[14], have no impact on −OH populations as long as Fe–O bonds are not displaced any longer than 2 Å (Figs. 4d and S7). Creating −OH in O sites of the Fh core should therefore require larger levels of disorder seen in poorly crystalline materials but not likely those of Fh nanoparticles displaying relatively more atomic order. Along the same vein, introducing vacancies at core Fe2 and Fe3 sites had no impact on −OH populations (Supplementary Fig. 7), although we note that concentrations can be effectively increased throughout the core with vacancies at

Fe1 sites (Fig. 4e). Our simulations thus show that −OH on spheroidal Fh surfaces are predominantly disposed as rows at edges of Fe1 sheets as corner (~2.7 –OH/nm$^2$) and edge (~2.0 –OH/nm$^2$) sites (Figs. 1 and 4f). We note that these populations fall in the lower range of densities previously resolved on crystalline faces by Hiemstra[35–38], still they readily account for metal ion and ligand adsorption densities reported in the literature[32,33,52].

Populations of μ−OH groups were strongly size-dependent in particles <~4 nm. They reached densities that were about ~3 times less than those of −OH in larger particles (Fig. 4g). While these sites are predominantly disposed as rows at edges of Fe1 sheets (Fig. 1), they are also exposed on basal faces (Fig. 4a), which are most clearly expressed in nanoparticles of up to ~6 nm in diameter. This result thus ties further the faint 3620 cm$^{-1}$ band (Fig. 2d) to μ-OH sites on the basal face of Fh. We also find that, unlike the case for −OH, vacancies in Fe2 and Fe3 sites increase μ−OH populations throughout the Fh core. As these populations become even larger with vacancies at Fe1 sites, we expect a greater variability of μ−OH populations throughout the core of poorly crystalline Fh. However, given their low ligand exchange capability[53,54] and proton affinity[43], we cannot expect enhanced reactivities from these sites alone.

## Inter-site interactions

**Inter-site interactions**. To understand how the distribution of surface OH groups impact inter-site interactions we turned to molecular dynamics simulations of single Fh nanoparticles with a surface depletion of Fe2 and Fe3 sites down to the first 4 Å. For these simulations, we removed all chemisorbed water molecules (−OH$_2$) from the surface, and retained full core Fe occupancies. This configuration was chosen to focus the discussion on the hydrogen bonding environment of OH groups of a hydroxylated Fh surface over a defect-free core. It thus emulates strongly dehydrated Fh surfaces, which were likely achieved in N$_2$(g) dried samples studied experimentally.

Our simulations retained the core crystallographic structure of Fh while mostly relaxing its hydroxylated surface (Fig. 5a–c). The surface sites in this family of simulation cells were spread over a ~0.6-nm region of the top portion of the particle, as imposed by the surface depletion scheme (Fig. 5a). This is the region whose structure is mostly affected by the relaxation of the hydroxylated shell. This relaxation can be especially appreciated by the radial distribution functions (Fig. 5c) of the 2-nm-wide nanoparticles where contributions from the surface shell are greatest (cf. Supplementary Figs. 8 and 9 for surface-specific radial distribution functions). The atomic profiles of this region (Fig. 5a) also reveal the relative depletion of μ$_3$-OH groups in the near surface region where the coordinatively unsaturated O sites are predominantly −OH and μ−OH. This also marks the zone in which hydrogen populations transition between a dominance of core μ$_3$−OH···O interactions (60–70% bonds per core OH) to those involving surface OH groups.

We find that Fh surfaces acquire an intricate network of hydrogen bonds that is predominantly built from donating μ−OH sites. These μ−OH sites donate hydrogen bonds up to ~$^1$/$_2$ of the available −OH (μ−OH···−OH) (Fig. 5d) and up to ~$^1$/$_5$ of μ−OH groups (μ−OH···μ−OH). The hydrogen bonding network also involves lateral μ−OH···OH−μ bonds taking place over both the dominant spherical surfaces and over the basal face. These interactions take proportionally more importance in the 2-nm particle, given the greater representation of the basal face (Fig. 4a). We also find that the higher curvature of the 2-nm nanoparticles strongly disfavours μ−OH···−OH hydrogen bonds (Fig. 5d) over the larger nanoparticles considered in this work. Because hydrogen bonding populations can strongly affect the proton affinity of −O

sites[50], we anticipate that these curvature-induced populations could play important roles in the acid–base chemistry of the smaller-size Fh nanoparticles. As such, consideration of these populations in future predictions of protonation constants, for example through extensions of the surface depletion model[35–38] or of atomistic simulations[55], can represent an important step to take in accounting for the size-dependent chemistry of Fh. Developments in this area will be highly beneficial in our pursuit for understanding the Fh surface reactivity, now that this study has established the identity and the likely dispositions of OH populations exposed at Fh nanoparticle surfaces.

By resolving the complex vibrational spectroscopic signals of Fh, this work offers new possibilities for tracking the catalytic reactivity of this nanomineral from the unique viewpoint of its OH groups. It immediately opens new possibilities for resolving the mechanisms through which Fh alters the fate of environmentally critical gases (e.g. CO$_2$, CH$_4$, NO$_x$, SO$_x$), as well as photo(electro)chemical transformation of organics in nature and technology. As these spectroscopic signals also persist under atmospheric humidity[47], this work also open new possibilities for exploring interfacial reactions and phase transformation in nanometrically thick water films at Fh surfaces. It may, additionally, be pertinent to investigate the possible roles that neutrally-charged basal faces play in the oriented aggregation of Fh in water. Future consideration of such possibilities, and especially based on our findings that hydrogen-bonded −OH and μ−OH groups dominate the surface speciation of Fh, represents some of the many anticipated implications that this work can have for understanding the surface chemistry of Fh nanoparticles in nature.

## Methods

**Fh synthesis and characterisation**. We synthesized 6-line Fh under N$_2$ (g) using the method of Schwertmann and Cornel[56]. To minimise atmospheric CO$_2$ contamination, we degassed all deionised water (18 MΩ·cm) used for synthesis and washing procedures by boiling then purging with N$_2$ (g) overnight. Material characterisation is reported in the Supplementary.

**Vibrational spectroscopy**. We collected vibrational spectra of N$_2$(g)-dried Fh particles by Fourier transform infrared spectroscopy. Following our previously established protocol[43], we altered the protonation levels of Fh surfaces as described in the Supplementary Methods. Centrifuged pastes of equilibrated minerals were thereafter transferred onto an attenuated total reflectance cell and dried to a thin film under N$_2$ (g) atmosphere. Spectra were collected with a Bruker Vertex 70/V Fourier transform infrared spectrometer, equipped with a DLaTGS.

We monitored the thermal stability of OH groups of Fh by temperature programmed desorption on dry samples by heating from 40 °C to 400 °C at a rate of 10 °C/min in a vacuum of <0.3 Pa. Dry Fh powders used for these experiments were pressed on a fine tungsten mesh (Unique wire weaving, 0.002″ mesh diameter) under a pressure of 5 tons, and inserted in a copper-heating shaft in direct contact with a K-type thermocouple. Transmission vibration spectra were collected in an optical chamber and water expelled from the sample was concomitantly detected by quadrupole mass spectrometry (Pfeiffer Vacuum, QMD 220 M2, PrismaPlus). The resulting spectra were analysed using a chemometric analysis[49] detailed in the Supplmentary Methods.

**Nanoparticle simulations**. We wrote a code to automatically generate and objectively analyse atomic populations of 280 different Fh nanoparticles. The nanoparticles generated for this work were 1–10 nm-wide spheroids cut from the crystallographic structure of Michel et al.[6]. The code then automatically modified the resulting nanoparticles with various combinations of the following strategies: (i) The surface depletion scheme was implemented by removing Fe2 and Fe3 sites within a preselected distance (e.g. 0, 2, 4, 6 Å) from the average position of top layer O atoms; (ii) Atomic displacements of random orientation were applied to all atoms with distances of up to 5 Å of their crystallographic position; (iii) Vacancies were randomly applied to up to 20% of only Fe2 and Fe3 sites (v$_{Fe2,Fe3}$) or all Fe atoms (v$_{Fe1,Fe2,Fe3}$). Next, we healed all undercoordinated atoms created as follows: (i) Free O and Fe atoms were removed from the structure; (ii) Fe sites lacking up to three coordinating O atoms were healed by adding first neighbour O atoms at their crystallographic positions, while those lacking more than three coordinating O atoms were removed; (iii) Dangling −O and μ−O were saturated by H atoms but the protonation state of core OH−1, O2, O3, O4 sites as in Michel et al.[6] were preserved as in the original crystallographic structure. Finally, charge neutrality was

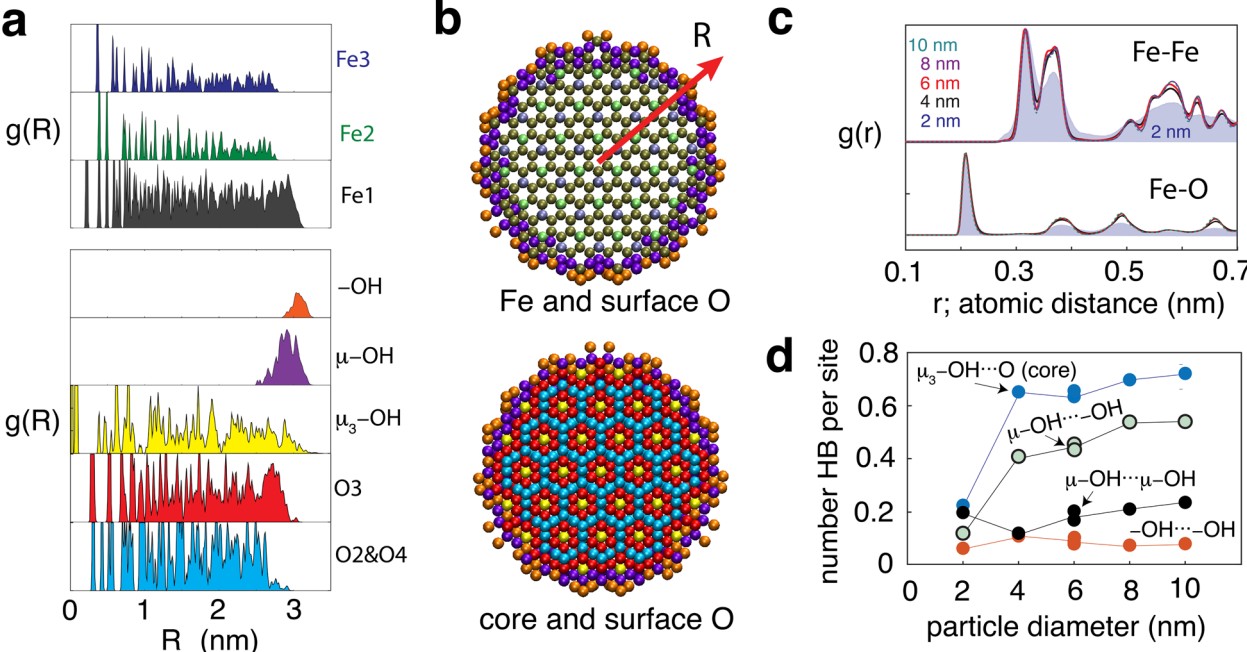

**Fig. 5 Molecular simulations of single dehydrated Fh nanoparticles. a** Example of the radial distribution function, g(R), of selected atoms of one 6-nm-wide particle with respect to the centre of mass (vector 'R' in **b**). The profiles show the depletion of Fe2 and Fe3, and the concentration of −OH and μ-OH sites at surfaces. **b** Cross section of a 2R = 6-nm-wide particle showing surface −OH (orange) and μ−OH (purple) sites, core Fe (Fe1, Fe2, Fe3) and core O (red, blue) and OH (yellow). **c** Radial distribution function, g(R), for Fe-Fe and Fe-O atomic distances (r) showing the particle size and surface OH population dependence on relative intensities. (cf. Supplementary Fig. 8 profiles for the top 0.6 nm, and Supplementary Fig. 9 for specific contributions from −OH groups). **d** Particle size dependence of hydrogen bonding populations. Note reproducibility of populations in simulations with six different surface distributions of −OH groups on the 6-nm-wide particles. These distributions resulted from the removal of randomly selected −OH₂ sites.

attained by ensured in all 280 Fh nanoparticles by protonating randomly selected −OH sites, which are the most proton-active sites of Fh.

All nanoparticle generation, treatment procedures and compositional analyses were performed using this code written in the computational environment of Matlab 9.1. The programme bookkeeps the identities of all Fe (Fe1, Fe2 and Fe3) and O atoms, as well as the identity of Fe sites to which all O are connected, and the coordination number ($−O$, $μ−O$, $μ_3−O$ and $μ_4−O$) of all O sites. For the latter, we use a cut-off distance of 2.4 Å for the Fe−O bond. The programme also determined nanoparticle volumes from the total number of oxygen, using the relationship $V_O = 10.8$ cm³/mol oxygen developed by Hiemstra and van Riemsdjik[38]. $V_O$ was then used to estimate mass densities. We found that this approach produced more reliable size-dependent mass densities than from volumes estimated by particle radius. For the same reason, we use the $V_O$ value to determine the total surface area of the particles, assuming a perfect sphere, to calculate all O (H) surface densities.

**Molecular dynamics.** We performed molecular dynamics of dehydrated Fh nanoparticles generated from our nanoparticle simulation code. We simulated neutrally-charged nanoparticles of 2 nm ($Fe_{90}O_{66}OH_{16}(−OH)_{47}(μ−OH)_{75}$) to 10 nm ($Fe_{17058}O_{21864}OH_{3428}(−OH)_{727}(μ−OH)_{3291}$) in diameter in an empty 20 × 20 × 20 nm³ box. The protonation state of −OH and μ−OH sites was chosen based on previous actual bond valence arguments[50,55], and charge neutrality was achieved by protonated randomly selected −OH groups. The resulting −OH₂ sites were then removed from the surface, thus exposing Lewis acid Fe³⁺ sites. Full details of the simulation procedures are reported in the SI Appendix.

## Data availability
Any relevant data are available from the authors upon reasonable request.

## Code availability
The suite of Matlab codes developed for this study can be accessed from the authors, and report of its use should cite this paper.

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

## Acknowledgements

This work was supported by the Swedish Research Council (2018–03808) to J.F.B. We thank the Swedish National Infrastructure for Computing (SNIC) for access to the computational facilities of the High Performance Computer Center North (HPC2N) of Umeå University. Open access funding provided by Umea University.

## Author contributions

J.-F.B. conceived, designed and supervised the project. X.S. performed all experimental work and assisted in the interpretation of the data. J.-F.B interpreted the experimental data, performed all computer programming and molecular simulation work, and wrote the paper.

## Competing interests

The authors declare no competing interests.
