## [Peer Review File · Communications Chemistry]

Reviewers' comments:

Reviewer #1 (Remarks to the Author):

The authors reported the distribution and characteristics of hydroxyl (OH) groups on ferrihydrite nanoparticles by infrared (IR) spectroscopy with change temperature and pH conditions aided by molecular dynamics (MD) simulations. They assigned several types of OH groups in the spectra based on their previous study for iron (Fe) (oxyhydr)oxides. The distribution of the OH groups in the nanoparticles were analyzed for various size of computationally generated nanoparticles by MD simulations. Through the analysis, reactive -OH group sites on ferrihydrite surfaces were identified.

The detail of surface structure of ferrihydrite is important to understand its strong scavenge ability of various elements including toxic elements such as arsenic. Thus, this research can be interested in various research fields especially for geological, environmental, and catalyst chemistries. They applied their knowledge obtained from previous research on the surface structure of Fe-(oxyhydr)oxides to that of the nanoparticles. This research can contribute to understanding of the reaction mechanism of sorption reactions taking place on the surface of ferrihydrite nanoparticles. I suggest minor revision before publishing the manuscript in several points as mention below.

1. Nanoparticle models were built by modifying cut ferrihydrite crystal structure with depletion of Fe atoms and adding protons (H⁺) randomly to terminal oxygen atoms (Fe-O) and OH groups for nanoparticles to be neutral. After that, the model structures were relaxed by classical MD simulations, but chemical reaction such as proton transfer never occur during the simulations. Thus, the characters such as the distribution of OH groups on the surface are almost determined by the model building process especially for the protonation process. It is better to confirm how the random protonation efficiently generate representative structures among the huge number of possible structures of ferrihydrite nanoparticles. Relating to the modeling process, the generality of the result for the analysis such as OH density and distribution shown in Figs. 4 and 5 should be shown, for instance, by comparison of the OH distribution among the same or similar size of nanoparticle models.

2. It is better to improve Figures and their captions to be more easily understood. For example, Figure 1: What is the difference between corner and edge-OH groups? Three O atoms of -OH are indicated by two arrows ('C' and 'E'). All these are, however, 'corner' -OH because edge is defined by two points. In addition, if the surface -OH groups are defined by 'C' and 'E', the center -OH group which is indicated by both arrows is doubly counted in the -OH density analysis decomposed by center and edge in Fig. 4F.

Figure 4C: The meaning of the star marks including their colors is not described well. The description of top 1-4 nm and how to calculate these values are also unclear even in the text. For example, for nearly 2 nm diameter particles, top 1 nm of the particle is almost similar to the whole particle. Thus, more detail explanations for the values can be necessary.

3. Though the manuscript, the word of "bulk" seems to be used in multiple meanings, e.g. crystalline (amorphous) in opposition to particulate for the mineral structure, whole particle or the core of particle (in Fig. 5B) opposite to the surface. Sometimes it is confusing. If possible, rewording can be help to reading correctly.

6. Lines 177-179: It is better to show the IR spectrum until ~3500 cm⁻¹ for Fig. 2D to confirm the absence of μ -3-OH vibrational mode.

7. Line 224: Fig. S2 can be Fig. S1.

8. Lines 301-343 and Figure 4: The details of the analysis are not described well, e.g. how to define and calculate -OH densities (nm⁻²), i.e. which area is used to calculate the value. These should be mentioned elsewhere.

9. Line 334: The "those -OH" can be "those μ -OH" to distinguish from -OH.
10. Lines 345-351: In this analysis all chemisorbed H₂O molecules are removed. What kind of experimental condition (temperature) does correspond to the simulation? This mention can be help to the interpretation of vibrational spectra in this current study.
11. Line 367: The word of "hydrated" should be remove because all H₂O molecules are removed for the bonding environment analysis.
12. Lines 367-369: The Fe-Fe and Fe-O distribution for 2 nm particle was mentioned as wider than larger size of nanoparticles based on the comparison of $g(r)$ in Fig. 5C. The contribution of Fe-Fe and Fe-O in core on $g(r)$ increase with increasing nanoparticle size, and Fh bulk (core) structure is retained during the MD simulations. As the result, $g(r)$ for the nanoparticles larger than 2 nm looks more like crystalline structure. If the flexibility of the surface is discussed based on the $g(r)$, $g(r)$ should be constructed by only Fe-Fe and Fe-O pairs in the surface shells (top ~ 0.6 nm region of nanoparticles) for each size of nanoparticles.
13. Line 385: The line "proton affinity is strongly affected by the number of hydrogen bonds" is not obvious. More detail explanation and addition of references are necessary.
14. Lines 467-469: Does the exposing Fe³⁺ create artificial Fe³⁺...OH/O interactions? If such Fe³⁺ sites confirmed in real conditions, some explanation can be necessary.

Reviewer #2 (Remarks to the Author):

Review Comments for COMMSCHEM-20-0062

I believe this manuscript should be accepted for publication after only minor revision.

The manuscript presents a detailed and direct analysis of the surface hydroxyl groups present on ferrihydrite (Fh) using both state of the art IR spectroscopic and molecular dynamics investigations. The major finding of the manuscript is that singly coordinated surface hydroxyl groups dominate the Fh surface, in a manner largely consistent with the surface depletion model of Hiemstra et al. The study is novel in that it provides direct evidence as to the general validity of that model. Given the importance of Fh in natural environments, the study should be of wide interest to primarily environmental scientists and geochemists interested in iron and trace element fate and transport near the earth's surface.

My primary criticism of the manuscript is that although its findings are largely consistent with the surface depletion model, it does not clearly distinguish any significant differences and/or discrepancies. For example, the extensive hydrogen bonding between groups identified is not part the original surface depletion model. There also appears to be some discrepancy in the site densities reported in this manuscript vs. the original surface depletion model. In any case, readers would benefit from a concise paragraph near the end of the manuscript which points out any significant differences/discrepancies vis a vis the surface depletion model.

Another question the authors might address concerns the potential impact of bulk water. The IR methods necessarily employed dried or semi-dried films of Fh whereas in natural environments Fh is more typically bathed in bulk aqueous solutions. So, the authors may wish to speculate how the presence of bulk water might modify their spectroscopic (and MD) findings.

Finally, the previous literature is adequately cited and utilized and the manuscript is clearly written. However, I have made note of a relatively few minor grammatical corrections on an annotated version of the manuscript and SI that should be made available to the authors.

Authors' response (blue) to Reviewers' comments (red)

Reviewer #1 (Remarks to the Author):

The authors reported the distribution and characteristics of hydroxyl (OH) groups on ferrihydrite nanoparticles by infrared (IR) spectroscopy with change temperature and pH conditions aided by molecular dynamics (MD) simulations. They assigned several types of OH groups in the spectra based on their previous study for iron (Fe) (oxyhydr)oxides. The distribution of the OH groups in the nanoparticles were analyzed for various size of computationally generated nanoparticles by MD simulations. Through the analysis, reactive – OH group sites on ferrihydrite surfaces were identified.

The detail of surface structure of ferrihydrite is important to understand its strong scavenge ability of various elements including toxic elements such as arsenic. Thus, this research can be interested in various research fields especially for geological, environmental, and catalyst chemistries. They applied their knowledge obtained from previous research on the surface structure of Fe-(oxyhydr)oxides to that of the nanoparticles. This research can contribute to understanding of the reaction mechanism of sorption reactions taking place on the surface of ferrihydrite nanoparticles.

I suggest minor revision before publishing the manuscript in several points as mention below.

We thank Reviewer #1 for a very thorough reading of this manuscript. This reviewer's comments were highly useful for improving this manuscript.

1. Nanoparticle models were built by modifying cut ferrihydrite crystal structure with depletion of Fe atoms and adding protons (H+) randomly to terminal oxygen atoms (Fe-O) and OH groups for nanoparticles to be neutral. After that, the model structures were relaxed by classical MD simulations, but chemical reaction such as proton transfer never occur during the simulations. Thus, the characters such as the distribution of OH groups on the surface are almost determined by the model building process especially for the protonation process. It is better to confirm how the random protonation efficiently generate representative structures among the huge number of possible structures of ferrihydrite nanoparticles. Relating to the modeling process, the generality of the result for the analysis such as OH density and distribution shown in Figs. 4 and 5 should be shown, for instance, by comparison of the OH distribution among the same or similar size of nanoparticle models.

The MD simulations that we presented in this study investigated only one set of randomly-selected protonation states for singly-coordinated groups of the ferrihydrite surface. The reviewer is concerned that a potential reorganization of the surface structure may prompt proton transfer between sites, as structure and protonation state can be interrelated. We share this concern especially in cases where structures are strongly altered. For example, had our MD simulations also involved particles with core Fe vacancies we would have had to search for a wide number of possible configurations, and this is where Monte Carlo simulations can be useful as in the work of Gilbert et al. ¹ and Funnell et al. ².

Our MD simulations however involve a relatively fixed crystalline core capped by a hydrated surface with a randomly-selected singly-coordinated OH and OH₂ sites. The concern raised by the review thus applies to the protonation state and positions of these and neighboring doubly- and triply-coordinate sites. It thus applies to our findings in the final section of this manuscript on 'Inter-site interactions', and thus to the radial distribution function of Fig. 5C and the hydrogen bonding environments of Fig. 5D. To respond to this comment, we begin by explaining how our previous work has shaped our thinking on choosing the protonation states of oxo groups, then we follow by reporting additional simulations of ferrihydrite supporting our approach.

Our previous MD work on many crystalline faces of (α , β , γ)-FeOOH and Fe₂O₃ minerals, and even on single Fe₂O₃ and Al(OH)₃ nanoparticles, guided our simulations on ferrihydrite. We estimate the protonation state of singly-, doubly- and triply-coordinated oxo groups with the method of Hiemstra and van Riemsdijk ³. This method predicts the protonation state from the crystalline Fe-O bond length and the likely hydrogen bond environments. Our MD simulations of these various faces allowed bond Fe-O bond lengths to contract/expand but none of these relaxed bonds have changed to such an extent that the protonation state of the oxo group should have changed according to the Brown-Altarmatt valence bond. In an earlier study by our group ⁴ we had even re-evaluated the protonation constants from the relaxed bonds, and that none of the relaxed Fe-O, O-H and O-H...H (H-bond) distances induced any major change in the surface speciation. We therefore believe that whatever change in surface structure that the ferrihydrite surface could experience should chiefly involve singly-coordinated groups. Protonation states of the core triply- and the surface doubly-coordinated groups should remain unchanged.

With these thoughts in mind, the situation that applies to ferrihydrite can be described as follows. We began by protonating all O sites according the method of Hiemstra and van Riemsdijk ³ for the starting structure. All singly- and doubly-coordinated groups were singly-protonated, while the crystalline core contains triply-coordinated OH groups and the core O₂ & O₄ groups remain unprotonated. Next, we achieved charge neutrality by adding one extra proton per randomly selected singly-coordinated -OH sites, which are all

located at the surface. The resulting chemisorbed water molecules were removed to perform simulations on dehydrated particle surfaces, such as those studied experimentally. As such, we stress that the core retained its crystalline structure throughout those simulations.

In this revised version of the manuscript we report the results of additional simulations on the 6 nm-wide nanoparticles containing 6 different randomly-protonated $-OH/-OH_2$ groups. The resulting hydrogen bond populations are highly reproducible, as shown in the new Figure 5D. Below is also a Table of results for this Reviewer.

Figure 5. Molecular simulations of neutrally-charged single dehydrated Fh nanoparticles from 2 to 10 nm in size ($Fe_{90}O_{66}OH_{16}(-OH)_{47}(\mu-OH)_{75}$ to $Fe_{17058}O_{21864}OH_{3428}(-OH)_{727}(\mu-OH)_{3291}$). These nanoparticles expose lower densities of $-OH$ than $\mu-OH$ because chemisorbed water molecules at $-OH_2$ sites were removed. (A) Example of the radial distribution function, $g(R)$, of selected atoms of one 6 nm wide particle with respect to the center of mass (vector ‘R’ in B). The profiles show the depletion of Fe2 and Fe3, and the concentration of $-OH$ and $\mu-OH$ sites at surfaces. (B) Cross section of a $2R = 6$ nm wide particle showing surface $-OH$ (orange) and $\mu-OH$ (purple) sites, core Fe (Fe1, Fe2, Fe3) and core O (red, blue) and OH (yellow). (C) Radial distribution function, $g(R)$, for Fe-Fe and Fe-O atomic distances (r) showing the particle size and surface OH population dependence on relative intensities. (*cf.* Fig. S8 profiles for the top 0.6 nm, and Fig. S9 for specific contributions from $-OH$ groups). (D) Particle size dependence of hydrogen bonding populations. Note reproducibility of populations in simulations with 6 different surface distributions of $-OH$ groups on the 6 nm-wide particles. These distributions resulted from the removal of randomly selected $-OH_2$ sites.

Table 1: Hydrogen Bonding Analysis for 5 repeated MD simulations for this revision for 6 nm-wide particles. (Results are reported in the revised Figure 5 D)

simulations	$\mu_3-OH \dots O_{core}$	$-OH \dots -OH$	$\mu-OH \dots \mu-OH$	$\mu-OH \dots -OH$
Version 1	0.63	0.09	0.17	0.43
Repeat 1	0.64	0.11	0.20	0.42
Repeat 2	0.64	0.11	0.19	0.45
Repeat 3	0.64	0.10	0.19	0.43
Repeat 4	0.63	0.09	0.20	0.46
Repeat 5	0.66	0.07	0.19	0.44

2. It is better to improve Figures and their captions to be more easily understood. For example, Figure 1: What is the difference between corner and edge-OH groups? Three O atoms of -OH are indicated by two arrows ('C' and 'E'). All these are, however, 'corner' -OH because edge is defined by two points. In addition, if the surface -OH groups are defined by 'C' and 'E', the center -OH group which is indicated by both arrows is doubly counted in the -OH density analysis decomposed by center and edge in Fig. 4F. Figure 4C: The meaning of the star marks including their colors is not described well. The description of top 1-4 nm and how to calculate these values are also unclear even in the text. For example, for nearly 2 nm diameter particles, top 1 nm of the particle is almost similar to the whole particle. Thus, more detail explanations for the values can be necessary.

We have re-evaluated the clarity of all figures and description of all figure captions of this paper:

Figure 1: The revised figure makes a clearer distinction between 'C' and 'E' sites. We assure that 'C' sites have not been counted twice in Fig. 4F that 'C' sites in the 1st version of this manuscript.

Figure 1. (A) Spherical Fh nanoparticle (8 nm diameter) cut from the structure of Michel *et al.*⁵ with a starting composition of $\text{Fe}_5\text{O}_8\text{H}$ ($\text{FeO}_{1.4}(\text{OH})_{0.2}$). A close-up of the selected area shows layers of octahedral (Fe1, Fe2) and tetrahedral (Fe3) iron atoms without (no SD) surface depletion of Fe2 and Fe3 sites. (B) Same close-up of selected area in A but with (SD) surface depletion of Fe2 and Fe3 sites. The face view of the selected area shows positions of Fe and O sites, including singly-coordinated corner ('C') and edge ('E') -OH, and doubly-coordinated μ -OH. Triply-coordinated μ_3 -O(H) groups chiefly belong to the core.

Figure 4C: The reviewer's question made us rethink of the utility of this portion of the graph. We have simplified the message in this revised version to simply show the effect of accounting for the top portion of a nanoparticle in accounting for XPS data. For simplicity, we limit our demonstration to one example only: the top 4 nm of a particles, which falls within a typical XPS analysis depth. To answer the reviewer's second question, these simulations do imply that the entire composition of a particle that is equal or less than 4 nm in diameter will also be retrieved by XPS. This is fully accounted for in this analysis. Below is our revised Figure 4:

Figure 4. (A) Simulated Fh nanoparticles at selected diameters, and their terminating hydroxo groups (orange = $-OH$; purple = $\mu-OH$; turquoise = μ_3-OH). The $\mu-OH$ groups of the basal faces are more clearly expressed in particles up to 6 nm. (B) Mass densities at given SDdis values and Fe vacancies. (C) OH/O ratios at SDdis=4 Å for entire and top 4 nm of the particles. This implies that OH/O ratios of particles less than 4 nm in diameters are from the entire particle. The dashed horizontal line marks the OH/O ratio of ~ 1.1 obtained by X-ray photoelectron spectroscopy (XPS). (D-G) Size dependence of OH density for (D) $-OH$ groups affected by surface depletion depth (SDdis) and atomic displacement of $\delta=3\text{Å}$, (E) $-OH$ groups affected by Fe vacancies at SDdis=4 Å, (F) breakdown of corner and edge sites (Fig. 1) of $-OH$ at SDdis=4 Å, and (G) $\mu-OH$ and μ_3-O densities at given values of SDdis. See Figs. S5-S7 for supporting results.

3. Though the manuscript, the word of “bulk” seems to be used in multiple meanings, e.g. crystalline (amorphous) in opposition to particulate for the mineral structure, whole particle or the core of particle (in Fig. 5B) opposite to the surface. Sometimes it is confusing. If possible, rewording can be help to reading correctly.

We agree that this is a source of confusion. We have therefore scanned the text for all references to the word “bulk”. We now always use the word “core” to refer to the “OH-poor” of Fh and “surface” for the topmost “OH-rich” region of Fh particles, as defined in lines 111-114:

The model envisions crystalline Fh nanoparticles composed of a defect-free core consisting of the low OH-bearing structure of Michel *et al.*⁵ (Fe_5O_8H), and of crystallographically-oriented OH/ H_2O -rich surfaces ($Fe_5O_8H + n H_2O$) depleted in Fe2 octahedral and Fe3 tetrahedral sites (Fig. 1)

All changes from ‘bulk’ to ‘core’ were highlighted in the text. Also when talking about $-OH$ and $\mu-OH$ groups created by Fe vacancies, we use the term ‘core’ to distinguish from their surface counterparts.

6. Lines 177-179: It is better to show the IR spectrum until $\sim 3500\text{ cm}^{-1}$ for Fig. 2D to confirm the absence of μ_3-OH vibrational mode.

The absence of μ_3-OH can already be appreciated in Fig. 2A where no peak is visible below 3600 cm^{-1} , and our reference bands for μ_3-OH on other minerals are shown in Fig. S1. We had considered the possibility of showing the entire $3500-3740$ region in an earlier version of Fig. 2D in the first version of this manuscript, but the figure did not show well the bands that are actually important for this work. We clarified this issue by working the text as follows:

Triply-coordinated μ_3-OH sites were not detected in the expected spectral range where they occur in other crystalline iron (oxyhydr)oxides^{6,7} minerals (Fig. 2A in the $3491-3578\text{ cm}^{-1}$ region; *cf.* Fig. S1 for reference spectra).

7. Line 224: Fig. S2 can be Fig. S1.

This is now changed. Thank you

8. Lines 301-343 and Figure 4: The details of the analysis are not described well, e.g. how to define and calculate $-OH$ densities (nm^{-2}), i.e. which area is used to calculate the value. These should be mentioned elsewhere.

The nanoparticle simulation section now describes this in greater detail:

All nanoparticle generation, treatment procedures and compositional analyses were performed using this code written in the computational environment of Matlab 9.1. The program bookkeeps the identities of all Fe (Fe1, Fe2, Fe3) and O atoms, as well as the identity of Fe sites to which all O are connected, and the coordination number ($-O$, $\mu-O$, μ_3-O , μ_4-O) of all O sites. For the latter, we use a cut-off distance of 2.4 Å for the Fe-O bond. The program also determined nanoparticle volumes from the total number of oxygen, using the relationship $V_O = 10.8 \text{ cm}^3/\text{mol oxygen}$ developed by Hiemstra and van Riemsdijk.⁸ V_O was then used to estimate mass densities. We found that this approach produced more reliable size-dependent mass densities than from volumes estimated by particle radius. For the same reason, we use the V_O value to determine the total surface area of the particles, assuming a perfect sphere, to calculate all O(H) surface densities.

9. Line 334: The “those $-OH$ ” can be “those $\mu-OH$ ” to distinguish from $-OH$.

We split the sentence two parts to improve clarity:

Populations of $\mu-OH$ groups were strongly size-dependent in particles less than ~ 4 nm. They reached densities that were about ~ 3 times to those of $-OH$ in larger particles (Fig. 4G).

10. Lines 345-351: In this analysis all chemisorbed H_2O molecules are removed. What kind of experimental condition (temperature) does correspond to the simulation? This mention can help to the interpretation of vibrational spectra in this current study.

This amounts to a fully dehydrated Fh surface, as likely achieved in our $N_2(g)$ -dried samples. We specified this further in the revised version through:

This configuration was chosen to focus the discussion on the hydrogen bonding environment of OH groups of a hydroxylated Fh surface over a defect-free core. It emulates strongly dehydrated Fh surfaces, which were likely achieved in $N_2(g)$ dried samples studied experimentally.

11. Line 367: The word of “hydrated” should be remove because all H_2O molecules are removed for the bonding environment analysis.

Thank you. We meant ‘hydroxylated’, which is now corrected.

12. Lines 367-369: The Fe-Fe and Fe-O distribution for 2 nm particle was mentioned as wider than larger size of nanoparticles based on the comparison of $g(r)$ in Fig. 5C. The contribution of Fe-Fe and Fe-O in core on $g(r)$ increase with increasing nanoparticle size, and Fh bulk (core) structure is retained during the MD simulations. As the result, $g(r)$ for the nanoparticles larger than 2 nm looks more like crystalline structure. If the flexibility of the surface is discussed based on the $g(r)$, $g(r)$ should be constructed by only Fe-Fe and Fe-O pairs in the surface shells (top ~ 0.6 nm region of nanoparticles) for each size of nanoparticles.

We entirely agree with the reviewer’s analysis of the size dependence of the $g(r)$ data. We indeed reported in Fig. 5D $g(r)$ values for all (core&surface) Fe-Fe and Fe-O shells. A breakdown of these values for all Fe1, Fe2 and Fe3 sites with $-OH$ sites was provided in Fig. S8 (now S9). We chose to report the pairs for both core and surface to facilitate comparison with experimental PDF values for future studies, and especially with simulations accounting for core Fe vacancies. Still, we understand the interest in extracting the $g(r)$ profiles for the topmost portion of the nanoparticles and we now provide these results in a new Fig. S8 of the revised Supporting Information section. This new analysis shows that the $g(r)$ profile of only the 2 nm-wide nanoparticles are affected by the surface composition. As this is not an important conclusion for this work, which is rather aimed at the direct identification and interactions of surface hydroxo groups, we relegated the results to the Supplementary Information section. We are hopeful that it will be of great utility for experimental PDF studies of size-dependent Fh nanoparticles:

Figure S8. Radial distribution function (RDF) for (A) Fe-Fe and (B) Fe-O atomic distances. Here we compare the values for all atoms (core&surface, as in Fig. 5C of main text) with those for only the top 0.6 nm of the particles. These results show that the RDF values for the combined core and surface of the 2 nm particles are dominated by the surface contributions, while those of the larger-sized particles are dominated by the core. Generated by Molecular Dynamics simulations of single Fh nanoparticles with diameters of 10, 20, 30 and 40 nm. $-\text{OH}$ groups are bound to only Fe1.

13. Line 385: The line “proton affinity is strongly affected by the number of hydrogen bonds” is not obvious. More detail explanation and addition of references are necessary.

Agreed. We refer to the Venema *et al.* paper³ to support this claim. This paper accounts for changes in the actual valence of O sites based on the number of donating and accepting hydrogen bonds, and shows that these changes correlate with important changes in proton affinity. As this manuscript is not about proton affinities *per se* we chose to refer to the Venema *et al.* paper as follows:

Because hydrogen bonding populations can strongly affect the proton affinity of $-\text{O}$ sites³, we anticipate that these curvature-induced populations could play important roles on the acid-base chemistry of the smaller-size Fh nanoparticles. As such, consideration of these populations in future predictions of protonation constants, for example through extensions of the surface depletion model⁸⁻¹¹ or of atomistic simulations⁵⁶, can represent an important step to take for accounting the size-dependent chemistry of Fh.^{3,4}

14. Lines 467-469: Does the exposing Fe^{3+} create artificial $\text{Fe}^{3+}\dots\text{OH}/\text{O}$ interactions? If such Fe^{3+} sites confirmed in real conditions, some explanation can be necessary.

These are the Lewis acid sites upon which H_2O sorbs under real conditions. We can experimentally detect these water molecules bound to these sites at 3690 cm^{-1} , as we have previously done on lepidocrocite, akaganéite and hematite. Removing these water molecules by exposure to dry gases or by heating exposes these Lewis acid sites on neutrally-charged surfaces (Analogous sites are widely studied in the catalytic alumina literatures with probes like CO and pyridine). We searched for deviations in Fe-O bond distributions that could perhaps be attributed to artificial interactions between O(H) groups and adjacent pentacoordinated Fe^{3+} surface sites. Our search however provided no evidence for such interactions, and other Fe-O bonds appear to be unaffected.

Reviewer #2 (Remarks to the Author):

Review Comments for COMMSCHEM-20-0062

I believe this manuscript should be accepted for publication after only minor revision.

The manuscript presents a detailed and direct analysis of the surface hydroxyl groups present on ferrihydrite (Fh) using both state of the art IR spectroscopic and molecular dynamics investigations. The major finding of the manuscript is that singly coordinated surface hydroxyl groups dominate the Fh surface, in a manner largely consistent with the surface depletion model of Hiemstra et al. The study is novel in that it provides direct evidence as to the general validity of that model. Given the importance of Fh in natural environments, the study should be of wide interest to primarily environmental scientists and geochemists interested in iron and trace element fate and transport near the earth's surface.

Thank you this very careful review our of work.

My primary criticism of the manuscript is that although its findings are largely consistent with the surface depletion model, it does not clearly distinguish any significant differences and/or discrepancies. For example, the extensive hydrogen bonding between groups identified is not part the original surface depletion model. There also appears to be some discrepancy in the site densities reported in this manuscript vs. the original surface depletion model. In any case, readers would benefit from a concise paragraph near the end of the manuscript which points out any significant differences/discrepancies vis a vis the surface depletion model.

We thank Reviewer #2 for raising this. It is true that the original surface depletion model does not consider hydrogen bonding environments, and we now suggest this first in the introduction:

“We note that these populations fall in the lower range of densities previously resolved on crystalline faces by Hiemstra⁸⁻¹¹, still they readily account for metal ion and ligand adsorption densities reported in the literature.¹²⁻¹⁴”

We also suggest this in the final section of the paper the following:

“Because hydrogen bonding populations can strongly affect the proton affinity of –O sites³, we anticipate that these curvature-induced populations could play important roles on the acid-base chemistry of the smaller-size Fh nanoparticles. As such, consideration of these populations in future predictions of protonation constants, for example through extensions of the surface depletion model⁸⁻¹¹ or of atomistic simulations⁵⁶, can represent an important step to take for accounting the size-dependent chemistry of Fh.^{3,4}”

Concerning comparisons with the original surface depletion model, the manuscript already contains lines highlighting salient differences. We begin by explaining that our spherical cuts are responsible for the low density of triply-coordinated groups:

Our simulations thus show that –OH on spheroidal Fh surfaces are predominantly disposed as rows at edges of Fe1 sheets as corner (~2.6 sites/nm²) and edge (~2.0 sites/nm²) sites (Figs. 1, 4F). We note that these populations fall in the lower range of densities previously resolved on crystalline faces by Hiemstra⁸⁻¹¹, still they readily account for metal ion and ligand adsorption densities reported in the literature.¹²⁻¹⁴

We show that the populations of -OH groups fall in the lower range of densities to those retrieved by Hiemstra on crystalline faces of Fh. We emphasize differences further in the introduction with the statement:

Line 131: “We support our findings with an exhaustive theoretical analysis of OH populations on Fh nanoparticles of spheroidal morphology that are more representative of real materials.. This becomes important as spherical cuts of crystallographic structures are likely to favor O(H) populations of lower coordination with underlying Fe sites than on flat crystalline surfaces, as in Hiemstra⁸⁻¹¹.”

Another question the authors might address concerns the potential impact of bulk water. The IR methods necessarily employed dried or semi-dried films of Fh whereas in natural environments Fh is more typically bathed in bulk aqueous solutions. So, the authors may wish to speculate how the presence of bulk water might modify their spectroscopic (and MD) findings.

This is a very important question, and represents the next step to be undertaken by our group. We already have simulations underway of Fh particles exposed to various loadings of surface and bulk water, and have monitored hydration data by FTIR. As this will be the object of a forthcoming paper we prefer keeping this manuscript focused on the fundamental aspects of OH identity in Fh.

Finally, the previous literature is adequately cited and utilized and the manuscript is clearly written. However, I have made note of a relatively few minor grammatical corrections on an annotated version of the manuscript and SI that should be made available to the authors.

All annotations made by this Reviewer were implemented in the revised version. There were also 2 queries:

1. **Methods:** “MD performed in absence of water molecules. The results might therefore not be representative of the behavior of Fh nanoparticles in aqueous environments”

MD simulations were to emulate the dry surfaces studied experimentally. In the Discussion we explain that this work can now lead to the study of hydration and water-driven phase conversion:

“As these spectroscopic signals also persist under atmospheric humidity^{47,55} also enables new work exploring interfacial reactions and phase transformation all within the confines of nanometrically-thick water films covering Fh surface. Future consideration of such possibilities represents some of the many anticipated implications that this work may have in understanding the surface chemistry of Fh nanoparticles in nature.”

2. **Figure S2:** “The BET surface area of dried Fh is known to be considerably smaller than that of Fh in solution. This should be acknowledged”

We agree. However, this study is focused on *dry* materials, so this point is not needed here. It will however be very important to raise it in subsequent work where we will look at hydration.

References

- 1 Gilbert, B. *et al.* A disordered nanoparticle model for 6-line ferrihydrite. *American Mineralogist* **98**, 1465-1476.
- 2 Funnell, N. P. *et al.* Nanocomposite structure of two-line ferrihydrite powder from total scattering. *Communications Chemistry* **3**, 22.
- 3 Venema, P., Hiemstra, T., Weidler, P. G. & van Riemsdijk, W. H. Intrinsic proton affinity of reactive surface groups of metal (hydr)oxides: application to iron (hydr)oxides. *Journal of Colloid and Interface Science* **198**, 282-295.
- 4 Boily, J. F. Water Structure and Hydrogen Bonding at Goethite/Water Interfaces: Implications for Proton Affinities. *Journal of Physical Chemistry C* **116**, 4714-4724.
- 5 Michel, F. M. *et al.* The structure of ferrihydrite, a nanocrystalline material. *Science* **316**, 1726-1729.
- 6 Song, X. & Boily, J.-F. Structural controls on OH site availability and reactivity at iron oxyhydroxide particle surfaces. *Phys. Chem. Chem. Phys.* **14**, 2579 - 2586.
- 7 Boily, J. F. *et al.* Thin Water Films at Multifaceted Hematite Particle Surfaces. *Langmuir* **31**, 13127-13137.
- 8 Hiemstra, T. & Van Riemsdijk, W. H. A surface structural model for ferrihydrite I: Sites related to primary charge, molar mass, and mass density. *Geochimica et Cosmochimica Acta* **73**, 4423-4436.
- 9 Hiemstra, T. Surface structure controlling nanoparticle behavior: magnetism of ferrihydrite, magnetite, and maghemite. *Environmental Science-Nano* **5**, 752-764.
- 10 Hiemstra, T. Formation, stability, and solubility of metal oxide nanoparticles: Surface entropy, enthalpy, and free energy of ferrihydrite. *Geochimica Et Cosmochimica Acta* **158**, 179-198.
- 11 Hiemstra, T. Surface and mineral structure of ferrihydrite. *Geochimica et Cosmochimica Acta* **105**, 316-325.
- 12 Hiemstra, T. Ferrihydrite interaction with silicate and competing oxyanions: Geometry and Hydrogen bonding of surface species. *Geochimica Et Cosmochimica Acta* **238**, 453-476.
- 13 Hiemstra, T. & Zhao, W. Reactivity of ferrihydrite and ferritin in relation to surface structure, size, and nanoparticle formation studied for phosphate and arsenate. *Environmental Science-Nano* **3**, 1265-1279.
- 14 Smith, K. F. *et al.* Plutonium(IV) Sorption during Ferrihydrite Nanoparticle Formation. *ACS Earth and Space Chemistry* **3**, 2437-2442.

REVIEWERS' COMMENTS:

Reviewer #1 (Remarks to the Author):

Authors improved their manuscript well and its descriptions became clearer. In addition, the authors explained carefully for my comments. Therefore, my questions at the first review were almost solved. However, the author's explanation and the modified caption for Fig. 4C seems to be somewhat wrong as mention below. After this point become clear, the manuscript can be acceptable for publication.

Lines 302-303 (the caption for Fig. 4C): The authors mentioned as follows, "OH/O ratios of particles less than 4 nm in diameter are from the entire particle." From my understanding, "top 4 nm" means outermost shell of a particle with 4 nm width. If the case, the OH/O ratio of top 4 nm for particles less than 8 nm may be same to that of whole particle. Thus, the explanation for Fig. 4C should be considered again.

Reviewer #2 (Remarks to the Author):

I have examined the revised manuscript and authors comments and in my opinion, the manuscript is now acceptable for publication in Communications Chemistry.

However, I did make note of a very few more grammatical corrections in an annotated version of the revised manuscript (attached below), that should be made available to the authors for their consideration.

Authors' response (blue) to Reviewers' comments (red)

Reviewer #1 (Remarks to the Author):

Authors improved their manuscript well and its descriptions became clearer. In addition, the authors explained carefully for my comments. Therefore, my questions at the first review were almost solved. However, the author's explanation and the modified caption for Fig. 4C seems to be somewhat wrong as mention below. After this point become clear, the manuscript can be acceptable for publication.

Lines 302-303 (the caption for Fig. 4C): The authors mentioned as follows, "OH/O ratios of particles less than 4 nm in diameter are from the entire particle." From my understanding, "top 4 nm" means outermost shell of a particle with 4 nm width. If the case, the OH/O ratio of top 4 nm for particles less than 8 nm may be same to that of whole particle. Thus, the explanation for Fig. 4C should be considered again.

The reviewer correctly pointed out this remaining inconsistency. We have now corrected this, as Fig. 4c was generated by showing the top 1 nm of the particle, not 4 nm. Fig. 4c now correctly reports '1 nm' and the figure caption was corrected accordingly, as follows:

"Mass densities at given SDdis values and Fe vacancies. (C) OH/O ratios at SDdis=4Å for entire and top 1 nm of the particles. The dashed horizontal line marks the OH/O ratio of ~1.1 obtained by X-ray photoelectron spectroscopy (XPS)."

Reviewer #2 (Remarks to the Author):

I have examined the revised manuscript and authors comments and in my opinion, the manuscript is now acceptable for publication in Communications Chemistry.

However, I did make note of a very few more grammatical corrections in an annotated version of the revised manuscript (attached below), that should be made available to the authors for their consideration.

We thank the reviewer for these corrections. All corrections were implemented in the revised version of this manuscript.